# 3D Steerable CNNs: Learning Rotationally Equivariant Features in Volumetric Data

**Maurice Weiler***
University of Amsterdam
m.weiler@uva.nl

**Mario Geiger***
EPFL
mario.geiger@epfl.ch

**Max Welling**
University of Amsterdam, CIFAR,
Qualcomm AI Research
m.welling@uva.nl

**Wouter Boomsma**
University of Copenhagen
wb@di.ku.dk

**Taco Cohen**
Qualcomm AI Research
taco.cohen@gmail.com

## Abstract

We present a convolutional network that is equivariant to rigid body motions. The model uses scalar-, vector-, and tensor fields over 3D Euclidean space to represent data, and equivariant convolutions to map between such representations. These SE(3)-equivariant convolutions utilize kernels which are parameterized as a linear combination of a complete steerable kernel basis, which is derived analytically in this paper. We prove that equivariant convolutions are the most general equivariant linear maps between fields over $\mathbb{R}^3$. Our experimental results confirm the effectiveness of 3D Steerable CNNs for the problem of amino acid propensity prediction and protein structure classification, both of which have inherent SE(3) symmetry.

## 1 Introduction

Increasingly, machine learning techniques are being applied in the natural sciences. Many problems in this domain, such as the analysis of protein structure, exhibit exact or approximate symmetries. It has long been understood that the equations that define a model or natural law should respect the symmetries of the system under study, and that knowledge of symmetries provides a powerful constraint on the space of admissible models. Indeed, in theoretical physics, this idea is enshrined as a fundamental principle, known as Einstein's principle of general covariance. Machine learning, which is, like physics, concerned with the induction of predictive models, is no different: our models must respect known symmetries in order to produce physically meaningful results.

A lot of recent work, reviewed in Sec. 2, has focused on the problem of developing equivariant networks, which respect some known symmetry. In this paper, we develop the theory of SE(3)-equivariant networks. This is far from trivial, because SE(3) is both non-commutative and non-compact. Nevertheless, at run-time, all that is required to make a 3D convolution equivariant using our method, is to parameterize the convolution kernel as a linear combination of pre-computed steerable basis kernels. Hence, the 3D Steerable CNN incorporates equivariance to symmetry transformations without deviating far from current engineering best practices.

The architectures presented here fall within the framework of Steerable G-CNNs [8, 10, 40, 45], which represent their input as fields over a homogeneous space ($\mathbb{R}^3$ in this case), and use steerable

Source code is available at https://github.com/mariogeiger/se3cnn.

filters [15, 37] to map between such representations. In this paper, the convolution kernel is modeled as a tensor field satisfying an equivariance constraint, from which steerable filters arise automatically.

We evaluate the 3D Steerable CNN on two challenging problems: prediction of amino acid preferences from atomic environments, and classification of protein structure. We show that a 3D Steerable CNN improves upon state of the art performance on the former task. For the latter task, we introduce a new and challenging dataset, and show that the 3D Steerable CNN consistently outperforms a strong CNN baseline over a wide range of trainingset sizes.

## 2   Related Work

There is a rapidly growing body of work on neural networks that are equivariant to some group of symmetries [3, 9, 10, 12, 19, 20, 28, 30–32, 36, 42, 46]. At a high level, these models can be categorized along two axes: the group of symmetries they are equivariant to, and the type of geometrical features they use [8]. The class of regular G-CNNs represents the input signal in terms of *scalar fields* on a group $G$ (e.g. $\mathrm{SE}(3)$) or homogeneous space $G/H$ (e.g. $\mathbb{R}^3 = \mathrm{SE}(3)/\mathrm{SO}(3)$) and maps between feature spaces of consecutive layers via group convolutions [9, 29]. Regular G-CNNs can be seen as a special case of steerable (or induced) G-CNNs which represent features in terms of *more general fields* over a homogeneous space [8, 10, 27, 30, 40]. The models described in this paper are of the steerable kind, since they use general fields over $\mathbb{R}^3$. These fields typically consist of multiple independently transforming geometrical quantities (vectors, tensors, etc.), and can thus be seen as a formalization of the idea of convolutional capsules [18, 34].

Regular 3D G-CNNs operating on voxelized data via group convolutions were proposed in [43, 44]. These architectures were shown to achieve superior data efficiency over conventional 3D CNNs in tasks like medical imaging and 3D model recognition. In contrast to 3D Steerable CNNs, both networks are equivariant to certain discrete rotations only.

The most closely related works achieving full $\mathrm{SE}(3)$ equivariance are the Tensor Field Network (TFN) [40] and the N-Body networks (NBNs) [26]. The main difference between 3D Steerable CNNs and both TFN and NBN is that the latter work on irregular point clouds, whereas our model operates on regular 3D grids. Point clouds are more general, but regular grids can be processed more efficiently on current hardware. The second difference is that whereas the TFN and NBN use Clebsch-Gordan coefficients to parameterize the network, we simply parameterize the convolution kernel as a linear combination of steerable basis filters. Clebsch-Gordan coefficient tensors have 6 indices, and depend on various phase and normalization conventions, making them tricky to work with. Our implementation requires only a very minimal change from the conventional 3D CNN. Specifically, we compute conventional 3D convolutions with filters that are a linear combination of pre-computed basis filters. Further, in contrast to TFN, we derive this filter basis directly from an equivariance constraint and can therefore prove its completeness.

The two dimensional analog of our work is the $\mathrm{SE}(2)$ equivariant harmonic network [45]. The harmonic network and 3D steerable CNN use features that transform under irreducible representations of $\mathrm{SO}(2)$ resp. $\mathrm{SO}(3)$, and use filters related to the circular resp. spherical harmonics.

$\mathrm{SE}(3)$ equivariant models were already investigated in classical computer vision and signal processing. In [33, 38], a spherical tensor algebra was utilized to expand signals in terms of spherical tensor fields. In contrast to 3D Steerable CNNs, this expansion is fixed and not learned. Similar approaches were used for detection and crossing preserving enhancement of fibrous structures in volumetric biomedical images [13, 21, 22].

## 3   Convolutional feature spaces as fields

A convolutional network produces a stack of $K_n$ feature maps $f_k$ in each layer $n$. In 3D, we can model the feature maps as (well-behaved) functions $f_k : \mathbb{R}^3 \to \mathbb{R}$. Written another way, we have a map $f : \mathbb{R}^3 \to \mathbb{R}^{K_n}$ that assigns to each position $x$ a feature vector $f(x)$ that lives in what we call the *fiber* $\mathbb{R}^{K_n}$ at $x$. In practice $f$ will have compact support, meaning that $f(x) = 0$ outside of some compact domain $\Omega \in \mathbb{R}^3$. We thus define the feature space $\mathcal{F}_n$ as the vector space of continuous maps from $\mathbb{R}^3$ to $\mathbb{R}^{K_n}$ with compact support.

In this paper, we impose additional structure on the fibers. Specifically, we assume the fiber consists of a number of geometrical quantities, such as scalars, vectors, and tensors, stacked into a single

$K_n$-dimensional vector. The assignment of such a geometrical quantity to each point in space is called a *field*. Thus, the feature spaces consist of a number of fields, each of which consists of a number of channels (dimensions).

Before deriving SE(3)-equivariant networks in Sec. 4 we discuss the transformation properties of fields and the kinds of fields we use in 3D Steerable CNNs.

## 3.1 Fields, Transformations and Disentangling

What makes a geometrical quantity (e.g. a vector) anything more than an arbitrary grouping of feature channels? The answer is that under rigid body motions, information flows within the channels of a single geometrical quantity, but not between different quantities. This idea is known as Weyl's principle, and has been proposed as a way of formalizing the notion of disentangling [6, 23].

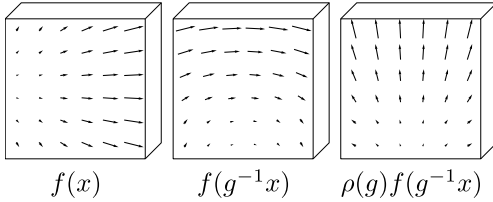

$$f(x) \qquad f(g^{-1}x) \qquad \rho(g)f(g^{-1}x)$$

Figure 1: To transform a vector field (L) by a 90° rotation $g$, first move each arrow to its new position (C), keeping its orientation the same, then rotate the vector itself (R). This is described by the induced representation $\pi = \mathrm{Ind}_{\mathrm{SO}(3)}^{\mathrm{SE}(2)} \rho$, where $\rho(g)$ is a $3 \times 3$ rotation matrix that mixes the three coordinate channels.

As an example, consider the three-dimensional vector field over $\mathbb{R}^3$, shown in Figure 1. At each point $x \in \mathbb{R}^3$ there is a vector $f(x)$ of dimension $K = 3$. If the field is translated by $t$, each vector $x - t$ would simply move to a new (translated) position $x$. When the field is rotated, however, two things happen: the vector at $r^{-1}x$ is moved to a new (rotated) position $x$, *and* each vector is itself rotated by a $3 \times 3$ rotation matrix $\rho(r)$. Thus, the rotation operator $\pi(r)$ for vector fields is defined as $[\pi(r)f](x) := \rho(r)f(r^{-1}x)$. Notice that in order to rotate this field, we need all three channels: we cannot rotate each channel independently, because $\rho$ introduces a functional dependency between them. For contrast, consider the common situation where in the input space we have an RGB image with $K = 3$ channels. Then $f(x) \in \mathbb{R}^3$, and the rotation can be described using the same formula $\rho(r)f(r^{-1}x)$ if we choose $\rho(r) = I_3$ to be the $3 \times 3$ identity matrix for all $r$. Since $\rho(r)$ is diagonal for all $r$, the channels do not get mixed, and so in geometrical terms, we would describe this feature space as consisting of three scalar fields, *not* a 3D vector field. The RGB channels each have an independent physical meaning, while the x and y coordinate channels of a vector do not.

The RGB and 3D-vector cases constitute two examples of fields, each one determined by a different choice of $\rho$. As one might guess, there is a one-to-one correspondence between the type of field and the type of transformation law (group representation) $\rho$. Hence, we can speak of a $\rho$-field.

So far, we have concentrated on the behaviour of a field under rotations and translations separately. A 3D rigid body motion $g \in \mathrm{SE}(3)$ can always be decomposed into a rotation $r \in \mathrm{SO}(3)$ and a translation $t \in \mathbb{R}^3$, written as $g = tr$. So the transformation law for a $\rho$-field is given by the formula

$$[\pi(tr)f](x) := \rho(r)f(r^{-1}(x - t)). \tag{1}$$

The map $\pi$ is known as the representation of SE(3) induced by the representation $\rho$ of SO(3), which is denoted by $\pi = \mathrm{Ind}_{\mathrm{SO}(3)}^{\mathrm{SE}(3)} \rho$. For more information on induced representations, see [5, 8, 17].

## 3.2 Irreducible SO(3) features

We have seen that there is a correspondence between the type of field and the type of inducing representation $\rho$, which describes the rotation behaviour of a single fiber. To get a better understanding of the space of possible fields, we will now define precisely what it means to be a representation of SO(3), and explain how any such representation can be constructed from elementary building blocks called irreducible representations.

A group representation $\rho$ assigns to each element in the group an invertible $n \times n$ matrix. Here $n$ is the dimension of the representation, which can be any positive integer (or even infinite). For $\rho$ to be called a representation of $G$, it has to satisfy $\rho(gg') = \rho(g)\rho(g')$, where $gg'$ denotes the composition of two transformations $g, g' \in G$, and $\rho(g)\rho(g')$ denotes matrix multiplication.

To make this more concrete, and to introduce the concept of an irreducible representation, we consider the classical example of a rank-2 tensor (i.e. matrix). A $3 \times 3$ matrix $A$ transforms under rotations as $A \mapsto R(r)AR(r)^T$, where $R(r)$ is the $3 \times 3$ rotation matrix representation of the abstract group element $r \in \mathrm{SO}(3)$. This can be written in matrix-vector form using the Kronecker / tensor product: $\mathrm{vec}(A) \mapsto [R(r) \otimes R(r)] \, \mathrm{vec}(A) \equiv \rho(r) \, \mathrm{vec}(A)$. This is a 9-dimensional representation of $\mathrm{SO}(3)$.

One can easily verify that the symmetric and anti-symmetric parts of $A$ remain symmetric respectively anti-symmetric under rotations. This splits $\mathbb{R}^{3 \times 3}$ into 6- and 3-dimensional linear subspaces that transform independently. According to Weyl's principle, these may be considered as distinct quantities, even if it is not immediately visible by looking at the coordinates $A_{ij}$. The 6-dimensional space can be further broken down, because scalar matrices $A_{ij} = \alpha \delta_{ij}$ (which are invariant under rotation) and traceless symmetric matrices also transform independently. Thus a rank-2 tensor decomposes into representations of dimension 1 (trace), 3 (anti-symmetric part), and 5 (traceless symmetric part). In representation-theoretic terms, we have reduced the 9-dimensional representation $\rho$ into irreducible representations of dimension $1, 3$ and $5$. We can write this as

$$\rho(r) = Q^{-1} \left[ \bigoplus_{l=0}^{2} D^l(r) \right] Q, \tag{2}$$

where we use $\bigoplus$ to denote the construction of a block-diagonal matrix with blocks $D^l(r)$, and $Q$ is a change of basis matrix that extracts the trace, symmetric-traceless and anti-symmetric parts of $A$.

More generally, it can be shown that any representation of $\mathrm{SO}(3)$ can be decomposed into irreducible representations of dimension $2l + 1$, for $l = 0, 1, 2, \ldots, \infty$. The irreducible representation acting on this $2l + 1$ dimensional space is known as the Wigner-D matrix of order $l$, denoted $D^l(r)$. Note that the Wigner-D matrix of order 4 is a representation of dimension 9, it has the same dimension as the representation $\rho$ acting on $A$ but these are two different representations.

Since any $\mathrm{SO}(3)$ representation can be decomposed into irreducibles, we only use irreducible features in our networks. This means that the feature vector $f(x)$ in layer $n$ is a stack of $F_n$ features $f^i(x) \in \mathbb{R}^{2l_i+1}$, so that $K_n = \sum_{i=1}^{F_n} 2l_{in} + 1$.

## 4 $\mathrm{SE}(3)$-Equivariant Networks

Our general approach to building $\mathrm{SE}(3)$-equivariant networks will be as follows: First, we will specify for each layer $n$ a linear transformation law $\pi_n(g) : \mathcal{F}_n \to \mathcal{F}_n$, which describes how the feature space $\mathcal{F}_n$ transforms under transformations of the input by $g \in \mathrm{SE}(3)$. Then, we will study the vector space $\mathrm{Hom}_{\mathrm{SE}(3)}(\mathcal{F}_n, \mathcal{F}_{n+1})$ of equivariant linear maps (intertwiners) $\Phi$ between adjacent feature spaces:

$$\mathrm{Hom}_{\mathrm{SE}(3)}(\mathcal{F}_n, \mathcal{F}_{n+1}) = \{ \Phi \in \mathrm{Hom}(\mathcal{F}_n, \mathcal{F}_{n+1}) \,|\, \Phi \pi_n(g) = \pi_{n+1}(g) \Phi, \;\; \forall g \in \mathrm{SE}(3) \} \tag{3}$$

Here $\mathrm{Hom}(\mathcal{F}_n, \mathcal{F}_{n+1})$ is the space of linear (not necessarily equivariant) maps from $\mathcal{F}_n$ to $\mathcal{F}_{n+1}$.

By finding a basis for the space of intertwiners and parameterizing $\Phi_n$ as a linear combination of basis maps, we can make sure that layer $n + 1$ transforms according to $\pi_{n+1}$ if layer $n$ transforms according to $\pi_n$, thus guaranteeing equivariance of the whole network by induction.

As explained in the previous section, fields transform according to induced representations [5, 8, 10, 17]. In this section we show that equivariant maps between induced representations of $\mathrm{SE}(3)$ can always be expressed as convolutions with equivariant / steerable filter banks. The space of equivariant filter banks turns out to be a linear subspace of the space of filter banks of a conventional 3D CNN. The filter banks of our network are expanded in terms of a basis of this subspace with parameters corresponding to expansion coefficients.

Sec. 4.1 derives the linear constraint on the kernel space for arbitrary induced representations. From Sec. 4.2 on we specialize to representations induced from irreducible representations of $\mathrm{SO}(3)$ and derive a basis of the equivariant kernel space for this choice analytically. Subsequent sections discuss choices of equivariant nonlinearities and the actual discretized implementation.

## 4.1 The Subspace of Equivariant Kernels

A continuous linear map between $\mathcal{F}_n$ and $\mathcal{F}_{n+1}$ can be written using a continuous kernel $\kappa$ with signature $\kappa : \mathbb{R}^3 \times \mathbb{R}^3 \to \mathbb{R}^{K_{n+1} \times K_n}$, as follows:

$$[\kappa \cdot f](x) = \int_{\mathbb{R}^3} \kappa(x, y) f(y) dy \tag{4}$$

**Lemma 1.** *The map $f \mapsto \kappa \cdot f$ is equivariant if and only if for all $g \in \mathrm{SE}(3)$,*

$$\kappa(gx, gy) = \rho_2(r)\kappa(x, y)\rho_1(r)^{-1}, \tag{5}$$

*Proof.* For this map to be equivariant, it must satisfy $\kappa \cdot [\pi_1(g)f] = \pi_2(g)[\kappa \cdot f]$. Expanding the left hand side of this constraint, using $g = tr$, and the substitution $y \mapsto gy$, we find:

$$\kappa \cdot [\pi_1(g)f](x) = \int_{\mathbb{R}^3} \kappa(x, gy)\rho_1(r)f(y)dy \tag{6}$$

For the right hand side,

$$\pi_2(g)[\kappa \cdot f](x) = \rho_2(r) \int_{\mathbb{R}^3} \kappa(g^{-1}x, y)f(y)dy. \tag{7}$$

Equating these, and using that the equality has to hold for arbitrary $f \in \mathcal{F}_n$, we conclude:

$$\rho_2(r)\kappa(g^{-1}x, y) = \kappa(x, gy)\rho_1(r). \tag{8}$$

Substitution of $x \mapsto gx$ and right-multiplication by $\rho_1(r)^{-1}$ yields the result. $\qquad\square$

**Theorem 2.** *A linear map from $\mathcal{F}_n$ to $\mathcal{F}_{n+1}$ is equivariant if and only if it is a cross-correlation with a rotation-steerable kernel.*

*Proof.* Lemma 1 implies that we can write $\kappa$ in terms of a one-argument kernel, since for $g = -x$ :

$$\kappa(x, y) = \kappa(0, y - x) \equiv \kappa(y - x). \tag{9}$$

Substituting this into Equation 4, we find

$$[\kappa \cdot f](x) = \int_{\mathbb{R}^3} \kappa(x, y)f(y)dy = \int_{\mathbb{R}^3} \kappa(y - x)f(y)dy = [\kappa \star f](x). \tag{10}$$

Cross-correlation is always translation-equivariant, but Eq. 5 still constrains $\kappa$ rotationally:

$$\kappa(rx) = \rho_2(r)\kappa(x)\rho_1(r)^{-1}. \tag{11}$$

A kernel satisfying this constraint is called *rotation-steerable*. $\qquad\square$

We note that $\kappa \star f$ (Eq. 10) is exactly the operation used in a conventional convolutional network, just written in an unconventional form, using a matrix-valued kernel ("propagator") $\kappa : \mathbb{R}^3 \to \mathbb{R}^{K_{n+1} \times K_n}$.

Since Eq. 11 is a linear constraint on the correlation kernel $\kappa$, the space of equivariant kernels (i.e. those satisfying Eq. 11) forms a vector space. We will now proceed to compute a basis for this space, so that we can parameterize the kernel as a linear combination of basis kernels.

## 4.2 Solving for the Equivariant Kernel Basis

As mentioned before, we assume that the $K_n$-dimensional feature vectors $f(x) = \oplus_i f^i(x)$ consist of irreducible features $f^i(x)$ of dimension $2\,l_{in} + 1$. In other words, the representation $\rho_n(r)$ that acts on fibers in layer $n$ is block-diagonal, with irreducible representation $D^{l_{in}}(r)$ as the $i$-th block. This implies that the kernel $\kappa : \mathbb{R}^3 \to \mathbb{R}^{K_{n+1} \times K_n}$ splits into blocks[1] $\kappa^{jl} : \mathbb{R}^3 \to \mathbb{R}^{(2j+1) \times (2l+1)}$ mapping between irreducible features. The blocks themselves are by Eq. 11 constrained to transform as

$$\kappa^{jl}(rx) = D^j(r)\kappa^{jl}(x)D^l(r)^{-1}. \tag{12}$$

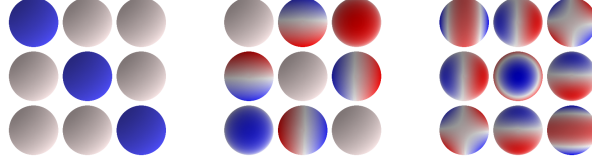

Figure 2: Angular part of the basis for the space of steerable kernels $\kappa^{jl}$ (for $j = l = 1$, i.e. 3D *vector* fields as input and output). From left to right we plot three $3 \times 3$ matrices, for $j - l \leq J \leq j + l$ i.e. $J = 0, 1, 2$. Each $3 \times 3$ matrix corresponds to one learnable parameter per radial basis function $\varphi^m$. A seasoned eye will see the identity, the curl ($\nabla \wedge$) and the gradient of the divergence ($\nabla \nabla \cdot$).

To bring this constraint into a more manageable form, we vectorize these kernel blocks to $\text{vec}(\kappa^{jl}(x))$, so that we can rewrite the constraint as a matrix-vector equation[2]

$$\text{vec}(\kappa^{jl}(rx)) = [D^j \otimes D^l](r)\,\text{vec}(\kappa^{jl}(x)), \tag{13}$$

where we used the orthogonality of $D^l$. The tensor product of representations is itself a representation, and hence can be decomposed into irreducible representations. For irreducible $\text{SO}(3)$ representations $D^j$ and $D^l$ of order $j$ and $l$ it is well known [17] that $D^j \otimes D^l$ can be decomposed in terms of $2\min(j, l) + 1$ irreducible representations of order[3] $|j - l| \leq J \leq j + l$. That is, we can find a change of basis matrix[4] $Q$ of shape $(2l + 1)(2j + 1) \times (2l + 1)(2j + 1)$ such that the representation becomes block diagonal:

$$[D^j \otimes D^l](r) = Q^T \left[ \bigoplus_{J=|j-l|}^{j+l} D^J(r) \right] Q \tag{14}$$

Thus, we can change the basis to $\eta^{jl}(x) := Q\,\text{vec}(\kappa^{jl}(x))$ such that constraint 12 becomes

$$\eta^{jl}(rx) = \left[ \bigoplus_{J=|j-l|}^{j+l} D^J(r) \right] \eta^{jl}(x). \tag{15}$$

The block diagonal form of the representation in this basis reveals that $\eta^{jl}$ decomposes into $2\min(j, l) + 1$ invariant subspaces of dimension $2J + 1$ with separated constraints:

$$\eta^{jl}(x) = \bigoplus_{J=|j-l|}^{j+l} \eta^{jl,J}(x), \qquad \eta^{jl,J}(rx) = D^J(r)\eta^{jl,J}(x) \tag{16}$$

This is a famous equation for which the *unique* and *complete* solution is well-known to be given by the spherical harmonics $Y^J(x) = (Y_{-J}^J(x), \ldots, Y_J^J(x)) \in \mathbb{R}^{2J+1}$. More specifically, since $x$ lives in $\mathbb{R}^3$ instead of the sphere, the constraint only restricts the angular part of $\eta^{jl}$ but leaves its radial part free. Therefore, the solutions are given by spherical harmonics modulated by an arbitrary continuous radial function $\varphi : \mathbb{R}^+ \to \mathbb{R}$ as $\eta^{jl,J}(x) = \varphi(\|x\|)Y^J(x/\|x\|)$.

To obtain a complete basis, we can choose a set of radial basis functions $\varphi^m : \mathbb{R}_+ \to \mathbb{R}$, and define kernel basis functions $\eta^{jl,Jm}(x) = \varphi^m(\|x\|)\,Y^J(x/\|x\|)$. Following [42], we choose a Gaussian radial shell $\varphi^m(\|x\|) = \exp\left(-\frac{1}{2}(\|x\| - m)^2/\sigma^2\right)$ in our implementation. The angular dependency at a fixed radius of the basis for $j = l = 1$ is shown in Figure 2.

By mapping each $\eta^{jl,Jm}$ back to the original basis via $Q^T$ and unvectorizing, we obtain a basis $\kappa^{jl,Jm}$ for the space of equivariant kernels between features of order $j$ and $l$. This basis is indexed by the radial index $m$ and frequency index $J$. In the forward pass, we linearly combine the basis kernels as $\kappa^{jl} = \sum_{Jm} w^{jl,Jm}\kappa^{jl,Jm}$ using learnable weights $w$, and stack them into a complete kernel $\kappa$, which is passed to a standard 3D convolution routine.

## 4.3 Equivariant Nonlinearities

In order for the whole network to be equivariant, every layer, including the nonlinearities, must be equivariant. In a regular G-CNN, any elementwise nonlinearity will be equivariant because the regular representation acts by permuting the activations. In a steerable G-CNN however, special equivariant nonlinearities are required.

Trivial irreducible features, corresponding to scalar fields, do not transform under rotations. So for these features we use conventional nonlinearities like ReLUs or sigmoids. For higher order features we considered tensor product nonlinearities [26] and norm nonlinearities [45], but settled on a novel gated nonlinearity. For each non-scalar irreducible feature $\kappa_n^i \star f_{n-1}(x) = f_n^i(x) \in \mathbb{R}^{2l_{in}+1}$ in layer $n$, we produce a scalar gate $\sigma(\gamma_n^i \star f_{n-1}(x))$, where $\sigma$ denotes the sigmoid function and $\gamma_n^i$ is another learnable rotation-steerable kernel. Then, we multiply the feature (a non-scalar field) by the gate (a scalar field): $f_n^i(x)\,\sigma(\gamma_n^i \star f_{n-1}(x))$. Since $\gamma_n^i \star f_{n-1}$ is a scalar field, $\sigma(\gamma_n^i \star f_{n-1})$ is a scalar field, and multiplying any feature by a scalar is equivariant. See Section 1.3 and Figure 1 in the Supplementary Material for details.

## 4.4 Discretized Implementation

In a computer implementation of $\mathrm{SE}(3)$ equivariant networks, we need to sample both the fields / feature maps and the kernel on a discrete sampling grid in $\mathbb{Z}^3$. Since this could introduce aliasing artifacts, care is required to make sure that high-frequency filters, corresponding to large values of $J$, are not sampled on a grid of low spatial resolution. This is particularly important for small radii since near the origin only a small number of pixels is covered per solid angle. In order to prevent aliasing we hence introduce a radially dependent angular frequency cutoff. Aliasing effect originating from the radial part of the kernel basis are counteracted by choosing a smooth Gaussian radial profile as described above. Below we describe how our implementation works in detail.

### 4.4.1 Kernel space precomputation

Before training, we compute basis kernels $\kappa^{jl,Jm}(x_i)$ sampled on a $s \times s \times s$ cubic grid of points $x_i \in \mathbb{Z}^3$, as follows. For each pair of output and input orders $j$ and $l$ we first sample spherical harmonics $Y^J, |j-l| \le J \le j+l$ in a radially independent manner in an array of shape $(2J+1) \times s \times s \times s$. Then, we transform the spherical harmonics back to the original basis by multiplying by $Q^J \in \mathbb{R}^{(2j+1)(2l+1)\times(2J+1)}$, consisting of $2J+1$ adjacent columns of $Q$, and unvectorize the resulting array to $\mathrm{unvec}(Q^J Y^J(x_i))$ which has shape $(2j+1) \times (2l+1) \times s \times s \times s$.

The matrix $Q$ itself could be expressed in terms of Clebsch-Gordan coefficients [17], but we find it easier to compute it by numerically solving Eq. 14.

The radial dependence is introduced by multiplying the cubes with each windowing function $\varphi^m$. We use integer means $m = 0, \ldots, \lfloor s/2 \rfloor$ and a fixed width of $\sigma = 0.6$ for the radial Gaussian windows.

Sampling high-order spherical harmonics will introduce aliasing effects, particularly near the origin. Hence, we introduce a radius-dependent bandlimit $J_{\max}^m$, and create basis functions only for $|j-l| \le J \le J_{\max}^m$. Each basis kernel is scaled to unit norm for effective signal propagation [42]. In total we get $B = \sum_{m=0}^{\lfloor s/2 \rfloor} \sum_{|j-l|}^{J_{\max}^m} 1 \le (\lfloor s/2 \rfloor + 1)(2\min(j,l)+1)$ basis kernels mapping between fields of order $j$ and $l$, and thus a basis array of shape $B \times (2j+1) \times (2l+1) \times s \times s \times s$.

### 4.4.2 Spatial dimension reduction

We found that the performance of the Steerable CNN models depends critically on the way of downsampling the fields. In particular, the standard procedure of downsampling via strided convolutions performed poorly compared to smoothing features maps before subsampling. We followed [1] and experiment with applying a low pass filtering before performing the downsampling step which can be implemented either via an additional strided convolution with a Gaussian kernel or via an average pooling. We observed significant improvements of the rotational equivariance by doing so. See Table 2 in the Supplementary Material for a comparison between performances with and without low pass filtering.

### 4.4.3 Forward pass

At training time, we linearly combine the basis kernels using learned weights, and stack them together into a full filter bank of shape $K_{n+1} \times K_n \times s \times s \times s$, which is used in a standard convolution routine. Once the network is trained, we can convert the network to a standard 3D CNN by linearly combining the basis kernels with the learned weights, and storing only the resulting filter bank.

# 5 Experiments

We performed several experiments to gauge the performance and data efficiency of our model.

## 5.1 Tetris

In order to confirm the equivariance of our model, we performed a variant of the Tetris experiments reported by [40]. We constructed a 4-layer 3D Steerable CNN and trained it to classify 8 kinds of Tetris blocks, stored as voxel grids, in a fixed orientation. Then we test on Tetris blocks rotated by random rotations in $SO(3)$. As expected, the 3D Steerable CNN generalizes over rotations and achieves $99\pm2\%$ accuracy on the test set. In contrast, a conventional CNN is not able to generalize over larger unseen rotations and gets a result of only $27\pm7\%$. For both networks we repeated the experiment over 17 runs.

## 5.2 3D model classification

Moving beyond the simple Tetris blocks, we next considered classification of more complex 3D objects. The SHREC17 task [35], which contains 51300 models of 3D shapes belonging to 55 classes (chair, table, light, oven, keyboard, etc), has a 'perturbed' category where images are arbitrarily rotated, making it a well-suited test case for our model. We converted the input into voxel grids of size 64x64x64, and used an architecture similar to the Tetris case, but with an increased number of layers (see Table 3 in the Supplementary Material). Although we have not done extensive fine-tuning on this dataset, we find our model to perform comparably to the current state of the art, see Figure 3 and Table 4 in the Supplementary Material.

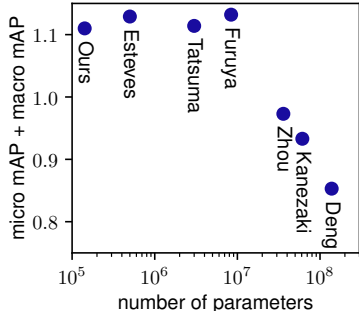

Figure 3: Shrec17 results[2, 7, 14, 16, 24, 35, 39]. Comparison of different architectures by number of parameters and score. See Table 4 in the Supplementary Material for all the details.

## 5.3 Visualization of the equivariance property

We made a movie to show the action of rotating the input on the internal fields. We found that the action are remarkably stable. A visualization is provided in `https://youtu.be/ENLJACPHSEA`.

## 5.4 Amino acid environments

Next, we considered the task of predicting amino acid preferences from the atomic environments, a problem which has been studied by several groups in the last year [4, 41]. Since physical forces are primarily a function of distance, one of the previous studies argued for the use of a concentric grid, investigated strategies for conducting convolutions on such grids, and reported substantial gains when using such convolutions over a standard 3D convolution in a regular grid ($0.56$ vs $0.50$ accuracy) [4].

Since the classification of molecular environments involves the recognition of particular interactions between atoms (e.g. hydrogen bonds), one would expect rotational equivariant convolutions to be more suitable for the extraction of relevant features. We tested this hypothesis by constructing the exact same network as used in the original study, merely replacing the conventional convolutional layers with equivalent 3D steerable convolutional layers. Since the latter use substantially fewer parameters per channel, we chose to use the same number of *fields* as the number of channels in the original model, which still only corresponds to roughly half the number of parameters (32.6M vs 61.1M (regular grid), and 75.3M (concentric representation)). Without any alterations to the model and using the same training procedure (apart from adjustment of learning rate and regularization factor), we obtained a test accuracy of $0.58$, substantially outperforming the conventional CNN on this task, and also providing an improvement over the state-of-the-art on this problem.

## 5.5 CATH: Protein structure classification

The molecular environments considered in the task above are oriented based on the protein backbone. Similar to standard images, this implies that the images have a natural orientation. For the final experiment, we wished to investigate the performance of our Steerable 3D convolutions on a problem domain with full rotational invariance, i.e. where the images have no inherent orientation. For this purpose, we consider the task of classifying the overall shape of protein structures.

We constructed a new data set, based on the CATH protein structure classification database [11], version 4.2 (see `http://cathdb.info/browse/tree`). The database is a classification hierarchy containing millions of experimentally determined protein domains at different levels of structural detail. For this experiment, we considered the CATH classification-level of "architecture", which splits proteins based on how protein secondary structure elements are organized in three dimensional space. Predicting the architecture from the raw protein structure thus poses a particularly challenging task for the model, which is required to not only detect the secondary structure elements at any orientation in the 3D volume, but also detect how these secondary structures orient themselves relative to one another. We limited ourselves to architectures with at least 500 proteins, which left us with 10 categories. For each of these, we balanced the data set so that all categories are represented by the same number of structures (711), also ensuring that no two proteins within the set have more than 40% sequence identity. See Supplementary Material for details. The new dataset is available at `https://github.com/wouterboomsma/cath_datasets`.

We first established a state-of-the-art baseline consisting of a conventional 3D CNN, by conducting a range of experiments with various architectures. We converged on a ResNet34-inspired architecture with half as many channels as the original, and global pooling at the end. The final model consists of $15, 878, 764$ parameters. For details on the experiments done to obtain the baseline, see Supplementary Material.

Following the same ResNet template, we then constructed a 3D Steerable network by replacing each layer by an equivariant version, keeping the number of 3D channels fixed. The channels are allocated such that there is an equal number of fields of order $l = 0, 1, 2, 3$ in each layer except the last, where we only used scalar fields ($l = 0$). This network contains only $143, 560$ parameters, more than a factor hundred less than the baseline.

We used the first seven of the ten splits for training, the eighth for validation and the last two for testing. The data set was augmented by randomly rotating the input proteins whenever they were presented to the model during training. Note that due to their rotational equivariance, 3D Steerable CNNs benefit only marginally from rotational data augmentation compared to the baseline CNN. We train the models for 100 epochs using the Adam optimizer [25], with an exponential learning rate decay of $0.94$ per epoch starting after an initial burn-in phase of $40$ epochs.

Despite having 100 times fewer parameters, a comparison between the accuracy on the test set shows a clear benefit to the 3D Steerable CNN on this dataset (Figure 4, leftmost value). We proceeded with an investigation of the dependency of this performance on the size of the dataset by considering reductions of the size of each training split in the dataset by increasing powers of two, maintaining the same network architecture but re-optimizing the regularization parameters of the networks. We found that the proposed model outperforms the baseline even when trained on a fraction of the training set size. The results further demonstrate the accuracy improvements across these reductions to be robust (Figure 4).

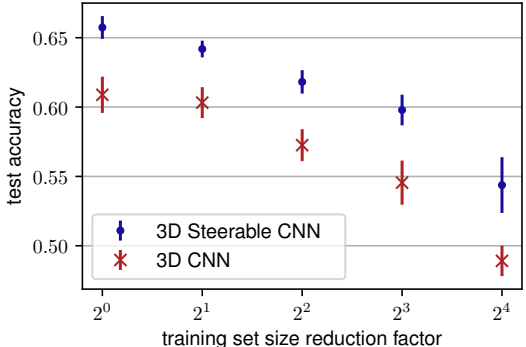

Figure 4: Accuracy on the CATH test set as a function of increasing reduction in training set size.

## 6   Conclusion

In this paper we have presented 3D Steerable CNNs, a class of $SE(3)$-equivariant networks which represents data in terms of various kinds of fields over $\mathbb{R}^3$. We have presented a comprehensive theory of 3D Steerable CNNs, and have proven that convolutions with $SO(3)$-steerable filters provide the most general way of mapping between fields in an equivariant manner, thus establishing $SE(3)$-equivariant networks as a universal class of architectures. 3D Steerable CNNs require only a minor adaptation to the code of a 3D CNN, and can be converted to a conventional 3D CNN after training. Our results show that 3D Steerable CNNs are indeed equivariant, and that they show excellent accuracy and data efficiency in amino acid propensity prediction and protein structure classification.

## Footnotes

* Equal Contribution. MG initiated the project, derived the kernel space constraint, wrote the first network implementation and ran the Shrec17 experiment. MW solved the kernel constraint analytically, designed the anti-aliased kernel sampling in discrete space and coded / ran many of the CATH experiments.

[1]For more details on the block structure see Sec. 2.7 of [10]

[2] vectorize correspond to flatten it in `numpy` and the tensor product correspond to `np.kron`

[3] There is a fascinating analogy with the quantum states of a two particle system for which the angular momentum states decompose in a similar fashion.

[4] $Q$ can be expressed in terms of Clebsch-Gordan coefficients, but here we only need to know it exists.

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
