[Supplementary Material]

# Supplementary material
# 3D Steerable CNNs: Learning Rotationally Equivariant Features in Volumetric Data

## 1 Design choices

### 1.1 Feature types and multiplicities

The choice of the types and multiplicities of the features is a hyperparameter of our network comparable to the choice of channels in a conventional CNN. As in the latter we follow the logic of doubling the number of multiplicities when downsampling the feature maps. The types and multiplicities of the network's input and output are prescribed by the problem to be solved. If one uses only scalar fields, then the kernels can only be isotropic, higher order representation allows more complex kernels. A more detailed investigation of the choice of these hyperparameters is left open for future work.

### 1.2 Normalization

We implemented an equivariant version of batch normalization [?]. For scalar fields, our implementation matches with the usual batch normalization. For the nonscalar fields we normalize them with the average of their norms:

$$f_i(x) \mapsto f_i(x) \left( \frac{1}{|\mathcal{B}|} \sum_{j \in \mathcal{B}} \frac{1}{V} \int dx \|f_j(x)\|^2 + \epsilon \right)^{-1/2} \tag{1}$$

where $\mathcal{B}$ is the batch and $i, j$ are the batch indices.

In order to reduce the memory consumption, we merged the batch normalization operation with the convolution

$$\kappa \star \underbrace{(Af + B)}_{BN} = (A\kappa) \star f + \kappa \star B.$$

### 1.3 Nonlinearities

The nonlinearities of an equivariant network need to be adapted to be equivariant themselves. Note that the domain and codomain of the nonlinearities might transform under different representations. We give an overview over the nonlinearities with which we experimented in the following paragraphs.

**Elementwise nonlinearities** Scalar features do not transform under rotations. As a consequence, they can be acted on by elementwise nonlinearities as in conventional CNNs. We chose $\mathrm{ReLU}$ nonlinearities for all scalar features except those which are used as gates (see below).

$$\begin{array}{ccc}
\mathcal{F}_n^{\text{scalar}} & \xrightarrow{\mathrm{Ind}_{\mathrm{SO}(3)}^{\mathrm{SE}(3)}[\mathrm{id}](g)} & \mathcal{F}_n^{\text{scalar}} \\
\Big\downarrow{\scriptstyle\text{ReLU}} & & \Big\downarrow{\scriptstyle\text{ReLU}} \\
\mathcal{F}_{n+1}^{\text{scalar}} & \xrightarrow[\mathrm{Ind}_{\mathrm{SO}(3)}^{\mathrm{SE}(3)}[\mathrm{id}](g)]{} & \mathcal{F}_{n+1}^{\text{scalar}}
\end{array}$$

**Norm nonlinearity**  The representations we are considering are all orthogonal and hence preserve the norm of feature vectors:

$$\|\rho(r)f(x)\| = f^T(x)\rho^T(r)\rho(r)f(x) = f^T(x)f(x) = \|f(x)\| \quad \forall r \in \mathrm{SO}(3),\ f \in \mathcal{F}$$

It follows that any nonlinearity applied to the norm of the feature commutes with the group transformation. Denoting a positive bias by $\beta \in \mathbb{R}_+$, we experimented with norm nonlinearites of the form

$$f(x) \ \mapsto \ \sigma_{\text{norm}}(f)(x) \ := \ \mathrm{ReLU}\left(\|f(x)\| - \beta\right) \frac{f(x)}{\|f(x)\|}.$$

Intuitively, the bias acts as a threshold on the norm of the feature vectors, setting small vectors to zero and preserving the orientation of large feature vectors. In practice, this kind of nonlinearity tended to converge slower than the gated nonlinearities, therefore we did not use them in our final experiments. This issue might be related to the problem of finding a suitable initialization of the learned biases for which we could not derive a proper scale. Norm nonlinearities were considered before in [8].

$$\begin{array}{ccc}
\mathcal{F}_n & \xrightarrow{\mathrm{Ind}_{\mathrm{SO}(3)}^{\mathrm{SE}(3)}[\rho](g)} & \mathcal{F}_n \\
\Big\downarrow{\scriptstyle\sigma_{\text{norm}}} & & \Big\downarrow{\scriptstyle\sigma_{\text{norm}}} \\
\mathcal{F}_{n+1} & \xrightarrow[\mathrm{Ind}_{\mathrm{SO}(3)}^{\mathrm{SE}(3)}[\rho](g)]{} & \mathcal{F}_{n+1}
\end{array}$$

**Tensor product nonlinearity**  The tensor product of two fields $f^1$ and $f^2$ is in index notation defined by

$$[f^1 \otimes f^2]_{\mu\nu}(x) = f^1_\mu(x)f^2_\nu(x).$$

This operation is nonlinear and equivariant and hence can be used in neural networks. We denote this nonlinearity by

$$\sigma_\otimes : \ \mathcal{F}_n \oplus \mathcal{F}_n \ \rightarrow \ \mathcal{F}_{n+1} := \mathcal{F}_n \otimes \mathcal{F}_n.$$

Note that the output of this operation transforms under the tensor product representation $\rho \otimes \rho$ of the input representations $\rho$. In our framework we could perform a change of basis $Q$ defined by $Q\rho \otimes \rho Q^{-1} = \bigoplus_j D^j$ to obtain features transforming under irreducible representations.

$$\begin{array}{ccc}
\mathcal{F}_n \oplus \mathcal{F}_n & \xrightarrow{\mathrm{Ind}_{\mathrm{SO}(3)}^{\mathrm{SE}(3)}[\rho \oplus \rho](g)} & \mathcal{F}_n \oplus \mathcal{F}_n \\
\Big\downarrow{\scriptstyle\sigma_\otimes} & & \Big\downarrow{\scriptstyle\sigma_\otimes} \\
\mathcal{F}_{n+1} = \mathcal{F}_n \otimes \mathcal{F}_n & \xrightarrow[\mathrm{Ind}_{\mathrm{SO}(3)}^{\mathrm{SE}(3)}[\rho \otimes \rho](g)]{} & \mathcal{F}_{n+1} = \mathcal{F}_n \otimes \mathcal{F}_n
\end{array}$$

Figure 1: A gated nonlinearity requires one extra scalar field (represented by gray circles with an $I$) per nonscalar output fields (represented by circles with a $\rho$). Specifically, the number of scalar output channels for the preceding convolution operator is increased by the number of features acted on by gated nonlinearities, and the extra scalar fields are computed in the same way as any other scalar field. We use sigmoid for the gate fields. In this picture, there is one scalar field in the output. It is activated with a ReLU.

**Gated nonlinearity** The gated nonlinearity acts on any feature vector by scaling it with a data dependent gate. We compute the gating scalars for each output feature via a sigmoid nonlinearity $\sigma : \mathcal{F}_n^{\text{scalar}} \to \mathcal{F}_n^{\text{scalar}}$ acting on an associated scalar feature. Figure 1 shows how the gated nonlinaritiy is coupled with the convolution operation. One can view the gated nonlinearity as a special case of the norm nonlinearity since it operates by changing the length of the feature vector. Simultaneously it can also be seen as a tensor product nonlinearity where one of the two fields as a scalar field. We found that the gated nonlinearities work in practice better than the the other options described above.

$$
\begin{array}{ccc}
\mathcal{F}_n^{\text{scalar}} \oplus \mathcal{F}_n & \xrightarrow{\text{Ind}_{\text{SO}(3)}^{\text{SE}(3)}[\text{id} \oplus \rho](g)} & \mathcal{F}_n^{\text{scalar}} \oplus \mathcal{F}_n \\
\downarrow{\sigma_{\text{gate}}} & & \downarrow{\sigma_{\text{gate}}} \\
\mathcal{F}_{n+1} & \xrightarrow{\text{Ind}_{\text{SO}(3)}^{\text{SE}(3)}[\rho](g)} & \mathcal{F}_{n+1}
\end{array}
$$

## 2 Reduced parameter cost of 3D Steerable CNNs

In the main paper, we demonstrated that the 3D Steerable CNN outperforms a conventional CNN despite having many fewer parameters. To ensure that the reduced number of parameters would not be an advantage also for the conventional CNN (due to overfitting with the high-capacity network), we trained a series of conventional CNNs with reduced number of filters in each layer (Figure 2). Note that the relative performance gain of our model increases dramatically if we restrict the conventional CNN to use the same number of parameters as the Steerable CNN.

Figure 2: Performance of our 3D Steerable CNN compared to a conventional 3D CNN with varying numbers of filters.

## 3 The Tetris experiment

The architecture used for the Tetris experiment has 4 hidden layers, the kernel size is 5 and the padding is 4. We didn't use batch normalization. Table 1 shows the multiplicities of the fields representations and the sizes of the fields. We compare with a regular CNN that has the same feature

map sizes. The CNN is like the SE3 network simply without the constraint of being equivariant for rotation. It has therefore much more parameters since its kernels are unconstrained. The SE3 network has 41k parameters and the CNN has 6M parameters.

|         | $l=0$ | $l=1$ | $l=2$ | $l=3$ | size   | CNN features |
|---------|-------|-------|-------|-------|--------|--------------|
| input   | 1     |       |       |       | $36^3$ | 1            |
| layer 1 | 4     | 4     | 4     | 1     | $40^3$ | 43           |
| layer 2 | 16    | 16    | 16    |       | $22^3$ | 144          |
| layer 3 | 32    | 16    | 16    |       | $13^3$ | 160          |
| layer 4 | 128   |       |       |       | $17^3$ | 128          |
| output  | 8     |       |       |       | 1      | 8            |

Table 1: Architecture of the network for the Tetris experiment. Between layer 1-2 and 2-3 there is a stride of 2. Between layer 4 and the output there is a global average pooling.

| low pass filter | disabled        | enabled         |
|-----------------|-----------------|-----------------|
| CNN             | $24\% \pm 4\%$  | $27\% \pm 7\%$  |
| SE3             | $36\% \pm 6\%$  | $99\% \pm 2\%$  |

Table 2: Test accuracy to classify rotated pieces of Tetris. Average and standard deviation over 17 runs.

## 4  3D Model classification

To find the model we ran 10 different models by changing depth, multiplicities, dropout, low pass filter or stride and two initialization method.

For this experiment we used a kernel size of 5 and a padding of 4. We used batch normalization. In this architecture we did't used the low pass filters. Table 3 shows the multiplicities of the fields representations and the sizes of the fields. This network has 142k parameters.

We converted the 3d models into voxels of size $64 \times 64 \times 64$ with the following code `https://github.com/mariogeiger/obj2voxel`.

Table 4 compares our results with results of the original competition and two other articles [2, 3].

|         | $l=0$ | $l=1$ | $l=2$ | size   |
|---------|-------|-------|-------|--------|
| input   | 1     |       |       | $64^3$ |
| layer 1 | 8     | 4     | 2     | $34^3$ |
| layer 2 | 8     | 4     | 2     | $38^3$ |
| layer 3 | 16    | 8     | 4     | $21^3$ |
| layer 4 | 16    | 8     | 4     | $25^3$ |
| layer 5 | 32    | 16    | 8     | $15^3$ |
| layer 6 | 32    | 16    | 8     | $19^3$ |
| layer 7 | 32    | 16    | 8     | $12^3$ |
| layer 8 | 512   |       |       | $16^3$ |
| output  | 55    |       |       | 1      |

Table 3: Architecture of the network for the 3D Model experiment. Where the size decrease we used a stride of 2. Between the last hidden layer and the output there is a global average pooling.

## 5  The CATH experiment

### 5.1  The data set

The protein structures used in the CATH study were simplified to include only $C_\alpha$ atoms (one atom per amino acid in the backbone), and placed at the center of a $50^3$vx grid, where each voxel spans 0.2 nm. The values of the voxels were set to the densities arising from placing a Gaussian at each atom position, with a standard deviation of half the voxel width. Since we limit ourselves to grids of size 5 nm, we exclude proteins which expand beyond a 5 nm sphere centered around their center of mass. This constraint is only violated by a small fraction of the original dataset, and thus constitutes no severe restriction.

| | micro | | | macro | | | total | | |
|---|---|---|---|---|---|---|---|---|---|
| | P@R | R@N | mAP | P@R | R@N | mAP | score | input size | params |
| Furuya [4] | **0.814** | 0.683 | 0.656 | **0.607** | 0.539 | **0.476** | **1.13** | $126 \times 10^3$ | 8.4M |
| Esteves [3] | 0.717 | 0.737 | 0.685 | 0.450 | 0.550 | 0.444 | **1.13** | $\mathbf{2 \times 64^2}$ | 0.5M |
| Tatsuma [7] | 0.705 | **0.769** | **0.696** | 0.424 | **0.563** | 0.418 | 1.11 | $38 \times 224^2$ | 3M |
| Ours | 0.704 | 0.706 | 0.661 | 0.490 | 0.549 | 0.449 | 1.11 | $1 \times 64^3$ | **142k** |
| Cohen [2] | 0.701 | 0.711 | 0.676 | - | - | - | - | $6 \times 128^2$ | 1.4M |
| Zhou [1] | 0.660 | 0.650 | 0.567 | 0.443 | 0.508 | 0.406 | 0.97 | $50 \times 224^2$ | 36M |
| Kanezaki [5] | 0.655 | 0.652 | 0.606 | 0.372 | 0.393 | 0.327 | 0.93 | - | 61M |
| Deng [6] | 0.418 | 0.717 | 0.540 | 0.122 | 0.667 | 0.339 | 0.85 | - | 138M |

Table 4: Results of the SHREC17 experiment.

For training purposes, we constructed a 10-fold split of the data. To rule out any overlap between the splits (in addition to the 40% homology reduction), we further introduce a constraint that any two members from different splits are guaranteed to originate from different categories at the "superfamily" level in the CATH hierarchy (the lowest level in the hierarchy), and all splits are guaranteed to have members from all 10 architectures. Further details about the data set are provided on the website (`https://github.com/wouterboomsma/cath_datasets`).

## 5.2 Establishing a state-of-the-art baseline

The baseline 3D CNN architecture for the CATH task was determined through a range of experiments, ultimately converging on a ResNet34-like architecture, with half the number of channels compared to the original implementation (but with an extra spatial dimension), and using a global pooling at the end to obtain translational invariance. After establishing the architecture, we conducted additional experiments to establish good values for the learning and drop-out rates (both in the linear and in the convolutional layers). We settled on a $0.01$ dropout rate in the convolutional layers, and L1 and L2 regularization values of $10^{-7}$. The final model consists of $15,878,764$ parameters.

## 5.3 Architecture details

Following the same ResNet template, we then constructed a 3D Steerable network, by replacing each layer with its equivariant equivalent. In contrast to the model architecture for the amino acid environment, we here opted for a minimal architecture, where we use exactly the same number of *3D channels* as in the baseline model, which leads to a model with the following block structure: $(2, 2, 2, 2), (((2, 2, 2, 2) \times 2) \times 3), (((4, 4, 4, 4) \times 2) \times 4), (((8, 8, 8, 8) \times 2) \times 6), (((16, 16, 16, 16) \times 2) \times 2 + ((256, 0, 0, 0))$. Here the 4-tuples represent fields of order $l = 0, 1, 2, 3$, respectively. The final block deviates slightly from the rest, since we wish to reduce to a scalar representation prior to the pooling. Optimal regularization settings were found to be a capsule-wide convolutional dropout rate of $0.1$, and L1 and L2 regularization values of $10^{-8.5}$. In this minimal setup, the model contains only $143,560$ parameters, more than a factor hundred less than the baseline.