[Reviews · NeurIPS 2018]

Reviewer 1



This paper proposes a convolutional network architecture where filters are defined as a linear combination of steerable kernel basis functions, such that the network is equivariant to rigid body motions. The paper's strength lie in the novelty of the proposed method for constructing equivariant filter banks from radial basis functions, as well as a novel equivariant nonlinearity. Unfortunately, the paper has a number of weaknesses that must be addressed for it to be acceptable for publication. First, the related work section should be expanded, since, although the paper cites many papers in the beginning, the section does not sufficiently explain the state of the field, contributions of prior works, and how the proposed method fits in and contributes upon them. Additionally, I suggest the authors reword their sentence on line 45 regarding the timing of a concurrent work, since it should not be a focus of this section. Secondly, I feel that section 3, which contains the main contributions of this paper, needs significant editing in order to be more clear, concise, and accessible to readers. I feel that certain more obvious preliminaries have been presented in an overly complicated way, introducing terminology that's inconsistent with previous works, or elaborating on the obvious while hurrying over the more complex material. For example, subsections 3.1 and 3.2 especially could be much shorter and to the point since they present very basic and well known fundamentals, whereas 3.3.1 and 3.4 could be explained with more detail and organization. In addition to this issue, while the authors explain properties of fields, transformations, equivariance, or how SO(3) can be decomposed, they do not do enough to tell the story of why these points are important to their overall problem. I would suggest that the authors try to write these sections so that readers can clearly understand the motivations that lead to each next step in the paper, and in such a way that the material is well grounded and more intuitively linked to their problem application. The authors also often explain particular choices, such as their novel equivariant nonlinearity, or precomputation details, without sufficiently explaining the alternatives or justifying the underlying reason for their decision. Sure, the proposed nonlinearity satisfies their desired criteria, but the authors should explain why and how they arrived at that nonlinearity among the many possible solutions. And what are it's advantages or disadvantages? Did the authors try alternative nonlinearities, which did not perform well in practice? This would be good to know, and similar questions would hold for other implementation details in the paper. Third, and most importantly, the experimental section of this paper is severely lacking. The first two experiments presented in 4.1 and 4.2 are not extensive or even analyzed in much detail, one experiment showing 100% accuracy on a toy dataset compared to a baseline which by construction will not perform well, and the other showing a modest improvement from 56% to 58% accuracy over a previous method. The authors themselves downplay the importance of these two experiments (e.g. line 268 - 269). This would be fine if followed by a stronger section, but for their third experiment they present an entrirely new dataset (the proposal and design of a new dataset could be an entire section in of itself with more rigorous presentation of the data properties and statistics) and compare against a single baseline and no other methods. The authors show that their method has heigher test accuracy on the new dataset than the baseline with much fewer parameters, but it's not obvious to me that fewer parameters is actually a disadvantage for this new dataset. It's very possible that the baseline is overly expressive for the amount of data available, and even simpler baselines might be more convincing. Most importantly though, the authors should have experiments that compare with contemporary methods to show the proposed methods strengths and weaknesses, as well as more extensive experiments showing the performance of their method under different implementation details. Overall, the authors presented an interesting new SE(3) equivariant convolutional network, but they need to first strengthen and focus their experimental section, and second work on the presentation and clarity of their method in section 3. Finally, I encourage the authors to elaborate on how their method relates to other recent works, and clearly emphasize the broader significance of their contributions. -------------------------------------------------------------------- After reading the rebuttal, I would first like to thank the authors for their efforts to address all of the concerns raised in the initial review. In particular, the described changes and additions to the experimental section have addressed my primary concerns with the original manuscript. The addition of arbitrary rotations to the Tetris experiment, the ShapeNet/Shrec17 experiment, clarifications of other experimental details, and the promised availability of code and dataset tools greatly strengthen the submission. I also appreciate the additional figure illustrating the effect of the number of parameters on performance. It demonstrates the authors' claims are substantiated by thorough experiments, and also shows interesting details, i.e. the rough order of magnitude of parameters at which the 3D CNN achieves diminishing returns in performance is around 10^5 vs 10^7, which otherwise would not have been known to the reader. Aside from the experimental section, I thank the authors for expanding their related works section and for their considerations in making the presentation of their method more accessible, and detailing their design decisions. I still think that the authors should take special care to choose accessible terminology, to not invoke complexity for its own sake, and to give appropriate attention to their various preliminaries. I understand that the authors' terminology is consistent with prior works [4] and [6], however I would still argue that the terminology used, even in those works, is not accessible to a broad audience. For example, readers familiar with spatial transformer networks, capsules, or problems of 2D and 3D point correspondence more generally, may already have a good intuition for transformations of vector fields, but would find terms like "fiber" unfamiliar and unnecessary. Please consider that [4] and [6] are only a sample of the field, and that related works such as Tensor Field Networks [24] use different terminology and cite other related bodies of work, such as PointNet. So, if the authors truly wish to address a broader audience, I would recommend that they consider readers from such perspectives. Overall, I feel that the authors have addressed many of the concerns raised with the paper's presentation and accessibility. Most importantly, they have strengthened their experimental section significantly, and based on these changes I would now vote for accepting this submission. Updated Score: 7

Reviewer 2



This paper is far from my area of expertise. Since in big conferences like NIPS it makes very little sense to avoid submitting a review in such circumstances, and especially that we are a few days before the review deadline, I am doing my best to provide a review here. Unfortunately, my review here is based on my general expertise in ML and NNs, but I am far from being an expert to the particular details of the problem here. Problem definition: the paper address the problem of designing NNs (particularly CNNs) that are equivariant with respect to a known symmetry, in this case these are SE(3)-equivariant networks. As the authors argue, this problem is far from trivial due to the mathematical properties of this symmetry group. The motivation for this problem is that in many natural sciences setups (such as those explored in the experimental setup of this paper) such properties are known to hold and if a machine learning model respects them, it is likely to better model the data. Importantly, the method proposed by the authors has modest run-time demands: it only needs to parameterize its filter kernels as a linear combination of pre-computed steerable basis kernels. This way "the 3D Steerable CNN incorporates equivariance to symmetry transformations without deviating far from current engineering best practices." The experimental setup is thorough and interesting and it provides strong indications that the ideas of the authors reflect a significant progress and can facilitate further meaningful research on the problem. Strength: I admire the scientific motivation of the authors and the fact that they address such a worthy problem, even if it is challenging and technically complex. The proposed method has appealing properties that make it feasible for computation and implementation. The experimental results are also thorough, interesting and at some points challenging. My intuition is that I wold have given this paper a higher score (at least 8) had I been more knowledgeable of its research problem and modeling techniques. I feel that with my current knowledge I cannot stand behind a higher score. After reading the response: I would like to thank the authors for their response. Most of the response does not refer to my review but it did help me understand the paper.

Reviewer 3



This paper proposes a method for modeling 3D convolutional layers that are SE(3)-steerable, i.e. equivariant under rigid motions. The authors develop a theoretical basis for their approach, and show that steerable filters require significantly fewer parameters than equivalent CNNs, since they are constrained to lie on a lower-dimensional manifold and can be represented in a basis for the latter. The paper is cleanly written and the experiments reasonable, although it would be good to see the performance of 3D steerable CNNs on, say, a version of ShapeNet with shapes randomly rotated, instead of a toy Tetris example. I'm also not happy that the Tetris example tests only 90 degree rotations: why not arbitrary rotations? I did not validate all the math. Assuming the math is correct, I think the method is a useful contribution which addresses the classical problem of designing features invariant to rigid transforms. I recommend acceptance, but would prefer to see some more results on synthetically rotated standard 3D datasets. == Update after reading rebuttal == I thank the authors for doing the experiments I suggested. The conclusions are promising. I continue to be in favor, again with the disclaimer that I have not verified all the math. It would help if someone (R1?) has done this or can do this, in which I case I would be even more strongly in favor.